# From Theory to Practice: Developing the MOBILE Intervention for Adults with Depression

**DOI:** 10.3390/healthcare13212667

**Published:** 2025-10-22

**Authors:** Shani Volovic-Shushan, Naomi Josman, Lior Ben Baruch, Adi Stern

**Affiliations:** 1Department of Occupational Therapy, Faculty of Social Welfare and Health Sciences, University of Haifa, Haifa 3498838, Israel; njosman@univ.haifa.ac.il (N.J.); liorbenbaruch24@gmail.com (L.B.B.); 2Shalvata Mental Health Center, Hod-Hasharon 4534708, Israel; 3Department of Occupational Therapy, Faculty of Health Sciences, Ben-Gurion University of the Negev, Beer Sheva 8410501, Israel; adister@bgu.ac.il

**Keywords:** complex intervention, occupational therapy, major depressive disorder, ecological momentary assessment, personalized intervention

## Abstract

**Background/Objectives:** Major depressive disorder (MDD) is often characterized by mood instability and occupational imbalance, leading to impaired functioning and reduced quality of life. Despite increasing recognition of occupational therapy’s (OT’s) role in mental health, few interventions comprehensively describe both their development process and their theoretical rationale. This article aims to present the systematic development of the mood–occupation balance reciprocal model (MOBILE) intervention for people with MDD, guided by the Medical Research Council (MRC) framework for complex intervention. **Methods:** Development followed three phases: (1) identifying the evidence base, (2) developing theory, and (3) modeling processes and outcomes. Theoretical foundations integrated occupation- and mood/function-based models with client-centered, lived-experience perspectives. A preliminary ecological momentary assessment (EMA) study on mood and daily function interplay further informed the intervention development. Input from focus groups with occupational therapists and individuals with lived experiences supported its iterative refinement. Following MRC guidelines, a small pilot study (N = 2) was conducted with inpatients diagnosed with MDD to assess the feasibility and acceptability of the intervention and EMA procedures. The pilot evaluated recruitment, adherence, engagement, and practicality of delivery within a clinical setting. **Results**: The pilot study, although it included only two inpatients and thus limits generalizability, demonstrated high adherence, engagement, and feasibility. The EMA protocol was well tolerated, leading to minor refinements that enhanced its clinical applicability. Stakeholders emphasized the program’s relevance, adaptability, and the value of personalized planning tools. **Conclusions**: The MOBILE intervention is delivered as an individualized program to enhance mood stability and daily functioning among inpatients with MDD. It provides a theoretically grounded, context-sensitive framework integrating personalized goal setting and strategy use with the construction of a balanced routine. **Implications:** This article provides a comprehensive account of development procedures to support future evaluation, implementation, and integration into OT mental health practice.

## 1. Introduction

This article aims to present the systematic development of a novel occupational therapy (OT) intervention, the mood–occupation balance reciprocal model (MOBILE) intervention for people with major depressive disorder (MDD), following the Medical Research Council (MRC) guidelines for complex interventions (see Figure 1) [1,2]. By providing a transparent account of the intervention’s conceptualization and modeling phase, this work intends to contribute to the advancement of evidence-based OT practice and serve as an example for the structured development of complex interventions. Recognition of the importance of OT in promoting function and participation among people with mental health conditions has been growing [3]. However, there remains a notable scarcity of published interventions in the field that detail the interventions’ effectiveness, underlying development processes, and theoretical frameworks guiding their design [4]. This gap limits the reproducibility of interventions and the advancement of evidence-based OT practice.

The MOBILE intervention is a personalized OT program aimed at enhancing mood stability and daily function among inpatients diagnosed with MDD. It addresses mood instability (MI) and occupational imbalance, which are both recognized as core challenges in MDD [5,6] that negatively affect quality of life (QoL) and resilience [7,8]. Its development was grounded in preliminary empirical research, OT theories, and established models of daily function and mood regulation, forming an integrative framework that links mood processes to engagement in meaningful occupations (see Section 2 of this article (Development of the MOBILE intervention) for further details).

The MRC guidelines define *complex interventions* as characterized by the number and interaction of components, the range of outcomes, the expertise and skills required by those delivering and receiving the intervention, and the need for flexible delivery to accommodate individual differences and contexts [2]. The MOBILE intervention addresses these complexities through its multi-component structure, integration of behavioral and contextual variables, and personalized approach emphasizing individualized goal-setting and strategy implementation. Through the incorporation of real-life data and EMA, the intervention is dynamically adapted to participants’ unique occupational patterns and contexts.

This article reports on the development phase of the MOBILE intervention, following the MRC’s structured approach. This framework emphasizes identifying the evidence base and relevant theories, as well as modeling intervention processes and expected outcomes prior to formal evaluation [2]. The specific objective of this paper is to detail how the MOBILE intervention was developed, from initial concept to early modeling, thus providing a foundation for its future evaluation, adaptation, and broader implementation in clinical and research contexts.

This article is structured as follows: Section 2 outlines the theoretical foundations and empirical evidence underlying the development of the MOBILE intervention. Section 3 presents the intervention’s structure and session content, detailing the phased implementation approach and integration of EMA. Section 4 describes the feasibility pilot study, including methodological considerations and findings on implementation in an inpatient setting. Section 5 discusses principal implications, limitations, and directions for future research and clinical practice.

## 2. Development of the MOBILE Intervention

The development process involved three core stages: identifying the existing evidence base, identifying and developing theory, and using a modelling process to refine the intervention’s components before embarking on a full-scale evaluation [1]. Considering the core elements (described in Figure 1), which advocate for dynamic, context-sensitive approaches [2], this process was iterative rather than linear.

### 2.1. Identifying the Existing Evidence Base

*Major depressive disorder* is a highly prevalent and disabling mood disorder, recognized as a leading cause of global disease burden [9]. It is characterized by a persistent (weeks to months) pattern of low mood, low self-esteem, and loss of interest or pleasure in daily activities, resulting in significant impairments across multiple domains of daily functioning [10]. Recent research has highlighted *mood instability*, defined as frequent and intense fluctuations in emotional state over hours or days, as a salient and clinically significant feature of MDD [11,12]. Notably, mood instability is part of a broader construct of mood dynamics, which also includes emotional inertia and emotional differentiation [6]. In this context, mood instability specifically represents both the intensity (variability) and temporal aspects (dependency) of emotional shifts over time—capturing how much an individual’s affect departs from its mean level as well as how rapidly it changes from one moment to the next [11,12]. Compared to nonclinical populations, individuals with MDD exhibit significantly higher mood instability levels with greater instability. These levels correlate with increased symptom severity, reduced QoL, and elevated risk of relapse and hospitalization [12].

Major depressive disorder adversely affects daily functioning across a broad range of life domains, including instrumental activities of daily living, social participation, productivity, and occupational roles [13]. Individuals with MDD demonstrate reduced occupational balance, characterized by limited engagement in meaningful activities, difficulty maintaining daily routines and schedules, and challenges attaining and sustaining occupational roles and identity. These disruptions are associated with lower QoL and decreased sense of personal resilience [7,14]. *Resilience*, defined as the ability to self-regulate and maintain thoughts, behaviors, or emotions and, thus, achieve a state of psychological well-being and everyday balance, closely intertwines with these outcomes. The relationship is reciprocal: resilience positively influences long-term functional outcomes and daily balance, whereas impairments in occupational performance and QoL further undermine resilience [14].

The current literature highlights a substantial gap in empirical evidence regarding OT interventions and their effectiveness in the mental health field, particularly for individuals with MDD [4,15]. Several interventions have been identified as effective in improving daily functioning in individuals with mental-health-related conditions. Notably, return-to-work interventions, such as the Redesigning Daily Occupations^®^ program [16], have demonstrated strong evidence for reducing depressive symptoms and facilitating successful reintegration into the workplace. In addition, there is limited but encouraging evidence supporting lifestyle interventions, such as the Balancing Everyday Life [17] program, that focus on promoting engagement in various daily activities, achieving occupational balance, and improving health habits. Time-use interventions have also shown promise for individuals with diverse mental health conditions. The Action Over Inertia [18] and Occupational Connections programs [19,20] are designed to increase participation in meaningful occupations using cognitive approaches, psychoeducation, enabling engagement, and narrative strategies.

Despite these examples, a notable absence of high-quality studies evaluating individualized, client-centered OT interventions tailored to the unique needs of people with MDD remains. In particular, interventions that account for emotional regulation—defined as the ability to recognize, understand, and effectively manage one’s emotions—and for occupation, as well as their reciprocal relationship and mutual influence, are lacking. Emotional regulation and mood dynamics are closely intertwined; challenges in regulating emotions can manifest as mood instability, while frequent mood fluctuations further complicate effective emotion regulation [6]. This research gap underscores the need to further investigate related theories and models and develop evidence-based OT practices.

### 2.2. Identifying and Developing Theory

According to the MRC’s guidelines for complex interventions (Figure 1), the next step in development is to identify and develop a theory. A key early task is to understand the change process expected from the intervention, drawing on existing theory and new research [1].

#### 2.2.1. Identifying Theory

The MOBILE intervention is grounded in a robust theoretical framework that integrates occupation and lifestyle-based models with mood- and function-based models. It also incorporates a client-centered, lived experience perspective. The following paragraphs describe the key theories and models underpinning the intervention.

Occupation and lifestyle-based models offer a comprehensive lens through which to understand individuals’ patterns of engagement in daily activities, as well as the challenges and limitations to participation that people with functional disabilities experience. These models emphasize the significance of occupational patterns, time use, and lifestyle balance as central determinants of health, well-being, and occupational balance.

The Time Use Model [21] focuses on how individuals allocate their time across various daily activities. It considers factors such as age, lifestyle, occupational variety, and the subjective sense of meaning derived from these occupations. This model uses methodologies like time-use diaries and observational techniques to systematically explore participation patterns, occupational balance, and lifestyle, offering insights into the individual’s daily structure and QoL.

The Lifestyle Balance Model [22] defines *lifestyle balance* as a satisfying, meaningful, and sustainable pattern of daily occupations that aligns with an individual’s values, skills, and interests. The model links lifestyle balance to positive outcomes, such as QoL and well-being, and emphasizes the importance of congruence between daily routines and personal circumstances. This model underscores the role of occupational patterns, participation, and time use in fostering health and life satisfaction.

Mood- and function-based models provide a theoretical framework for understanding how individuals regulate their internal states and daily functioning in response to the demands of everyday life. Central to these models is the concept of *homeostasis*, the internal self-regulation processes that maintain stability, balance, and harmony within an organism [23]. Although homeostasis has traditionally been discussed in biological and physiological contexts, these models extend the concept to encompass behavioral and psychological adaptation in daily activities.

The Daily Function Homeostasis framework [24] offers a structured approach to understanding and mapping daily functioning across the lifespan. This model encourages individuals to systematically identify and reflect on their functional characteristics using guided questions (e.g., who, what, where, when, how, and why) to build a comprehensive profile of their daily activities and participation. By illuminating strengths and areas for improvement, the framework supports the development of self-awareness, insight, and targeted strategies for enhancing control, satisfaction, and overall well-being.

The concept of *mood homeostasis* explores the relationship between occupational balance and mood regulation. It is defined as an individual’s capacity to regulate and balance their mood by selecting and engaging in daily activities [25]. This model incorporates the hedonic flexibility principle, which posits that individuals tend to choose activities based on their current mood state. Specifically, they prefer mood-enhancing activities when feeling low and accept tasks that are less rewarding when their mood is more positive [26]. The dynamic interplay between mood and activity selection highlights the importance of adaptive strategies in maintaining emotional well-being and occupational balance.

Client-centered and lived experience perspectives are central to OT practice. These approaches emphasize collaboration between the therapist and client, ensuring that interventions are tailored to the individual’s unique needs, values, and goals [27]. In a client-centered perspective, therapists recognize clients as experts in their own experiences and agents in their daily lives and challenges. Integrating the lived experience perspective allows a deeper understanding of the client’s context and personal meaning, resulting in relevant and unique interventions that reflect real daily life experiences and challenges. Ultimately, this perspective fosters autonomy, shared decision-making, and meaningful engagement in the therapeutic process [28].

#### 2.2.2. Developing Theory

The next step after identifying the relevant theoretical framework, according to the guidelines, is to develop a theory by integrating the intervention’s key components to define a process of change. We conducted a preliminary study examining the dynamic relationship between these variables in individuals with MDD to support developing a robust theoretical framework that integrates the core concepts of mood instability and daily function [29]. This descriptive comparative study utilized ecological momentary assessment (EMA) to gather real-life data on mood and daily functioning. Individuals with MDD exhibited significantly higher mood instability, reduced occupational balance, less participation in daily activities, and less uniform activity patterns compared to controls. Higher mood instability was associated with greater depression severity, lower QoL, and decreased occupational balance. These findings underscored the complex interplay between mood instability and functional impairment in MDD, emphasizing the value of real-time mood and activity monitoring for understanding the lived experiences of individuals with MDD. The reciprocal relationships identified in this study [29] provide preliminary evidence for a shared balance-based clinical mechanism that forms the basis of the MOBILE model intervention in MDD (Figure 2). This model offers an integrative explanation for the daily experiences of individuals with MDD.

The MOBILE model is a dynamic framework integrating three core components: depression severity, mood instability, and occupation. Each component operates along a continuum from balance to imbalance (e.g., mood instability ranges from emotional instability to stability, daily function and occupation range from dysfunctional to adaptive patterns, and depression severity ranges from acute to remitted states). These components exhibit reciprocal relationships: depression severity exacerbates mood instability and disrupts occupational balance and daily function; impaired functioning amplifies mood instability and perpetuates depressive symptoms. These interactions ultimately affect participation in daily activities, QoL, and resilience.

The model emphasizes that therapeutic interventions must address all three components simultaneously because improvements in one component (e.g., stabilizing mood through activity modulation) can positively influence the entire system. By framing MDD challenges through this balance-based lens, the MOBILE model provides clinicians with a structured approach to identify personalized treatment targets while acknowledging the ecological complexity of depressive experiences.

### 2.3. Modeling Process and Outcomes

The MOBILE model served as the foundation for designing the MOBILE intervention. In the first stage, the four authors, experienced occupational therapists specializing in mental health, collaborated to build the intervention’s content. The content included a structured manual, delivery methods, clinical tools, and expected outcomes. Once we established the initial draft, we assessed content validity through a focus group with 13 expert occupational therapists otherwise unconnected to the study. The experts received a detailed overview of the study’s objectives and the intervention protocol. The review process included 20 questions on a Likert scale from 1 (not at all) to 5 (very much) and open-ended questions for qualitative feedback and suggested revisions or additions. Through this process, the focus group helped fully define the change process and clarify each intervention component’s theoretical foundations and intended outcomes.

Additionally, we collected data from six experts by experience—individuals with lived experiences of MDD who participated in the preliminary study. After we explained the intervention’s aims and structure, these participants provided their perspectives on the program’s content, format, and perceived feasibility using a structured questionnaire. The questionnaire comprised eight Likert-scale items and open-ended questions for qualitative feedback and suggestions for revision or addition. This parallel process ensured that both professional and experiential perspectives informed the ongoing refinement of the intervention. Table 1 summarizes the main topics addressed in the focus group and the questionnaires.

#### Stakeholder Perspectives and Intervention Refinement

Feedback from the occupational therapist focus group and the questionnaires completed by experts by experience (individuals with lived experience of MDD) underwent systematic qualitative and quantitative analysis to guide the adaptation and refinement of intervention components and expected outcomes. Quantitative ratings from the occupational therapists were analyzed using the intra-class correlation coefficient (ICC; α = 0.775). They indicated a good level of agreement among raters regarding the clarity, structure, content, and implementation of the intervention protocol.

The qualitative feedback underscored the importance of ensuring the intervention’s relevance to real-life situations. It showed a clear need to extend the intervention’s applicability beyond inpatient settings to diverse care environments, including acute care, outpatient clinics, participants’ homes, and community housing. The occupational therapists emphasized the value of facilitating strategy use and raising participant awareness of cognitive and emotional states while keeping instructions straightforward and adaptable. They also highlighted the need to adjust program demands responsively to match the participants’ current states.

The input from experts by experience was also integral to the refinement of the MOBILE protocol. Their feedback emphasized the importance of practical relevance, flexibility, and clarity in program materials. They provided unique recommendations focused on real-life applicability, simplification of tasks, and adaptation for diverse environments, which differed from the professionals’ emphasis on theory and structure. Several concrete protocol changes resulted, including the introduction of additional practical examples, expanded diary formats, and reduced cognitive burden during session activities.

Regarding the intervention’s strengths, both the occupational therapists and the experts by experience particularly noted the accessibility and organization of the forms and materials; the ongoing monitoring of progress, which allowed basing information on current experiences rather than retrospective memory; the use of visual tools to illustrate daily functioning and mood; and the weekly tailoring of the functional program to each participant’s current condition.

We revised and enriched the protocol with additional examples and clarifications Based on this comprehensive feedback. To support effective transfer and generalization, we incorporated the multicontext approach [30] (pp. 79–89) as a foundational method to guide the intervention and its tools.

The *multicontext approach* is a metacognitive, strategy-based intervention designed to optimize functional cognition by promoting self-awareness, strategy use, and the ability to transfer skills across a wide range of everyday activities. In this context, *transfer* refers to applying learned strategies from one task or situation to a similar or related one. *Generalization* extends this concept by describing the use of those strategies in tasks or situations that are more dissimilar or in natural, everyday environments [30] (pp. 108–120). Through this process, participants are encouraged to monitor their own performance, recognize challenges, and adapt their strategies as needed.

Additional outcomes of the modelling process included refining how the key theories and models underlying the intervention manifest in the MOBILE intervention’s components. Table 2 summarizes these outcomes.

Finally, per Step 3 of the MRC development process, we refined and specified the expected behaviors to clearly explain the change process and the anticipated improvements in participation. Specifically, the goals for intervention participants are to identify patterns of daily occupations and mood and to understand the reciprocal relationship between them, thereby increasing awareness of how occupational engagement and emotional states influence one another. In addition, participants are encouraged to learn strategies that promote occupational balance and emotional regulation, supporting balanced participation and greater mood stability in daily life. Finally, they are guided to apply these strategies to build and sustain a balanced daily routine, promoting the formation and generalization of habits that support ongoing participation and overall well-being.

## 3. The MOBILE Intervention Structure and Content

The MOBILE intervention is delivered as an individual program of eight 45 min structured sessions (twice weekly for 4 weeks). It is intended primarily for inpatients but can be adapted for outpatient or community settings. The intervention is organized into four sequential phases, each with specific objectives and content.

A central feature of MOBILE intervention is the systematic integration of EMA across all four phases. Throughout the intervention process, EMA enables real-time monitoring of mood, daily routines, and participation, supporting precise detection of dynamic changes and providing immediate, personalized feedback. This ongoing data stream strengthens clinical decision-making and allows for tailored intervention by informing both therapists and participants about current functional status and progress.

Specifically, EMA data are used to build a personalized functional profile for each participant, identify individual patterns of occupational engagement and mood fluctuation, and monitor weekly progress in functional activities. These insights guide the customization of session content and daily planning to match participants’ evolving experiences and needs, ensuring each session remains relevant and responsive.

Phase 1 focuses on assessing mood and function (participation and occupational balance) to create a personalized profile and set individual goals. Participants review their mood tracking and occupational balance results, map daily routines, and collaboratively identify functional goals that guide the intervention. This phase introduces core concepts such as occupational balance, mood instability, and mood regulation, emphasizing the reciprocal relationship between mood and daily activities. It combines psychoeducation, personal reflection, and diary use to monitor mood and participation. To illustrate this process, Figure 3 presents an EMA-based graph from a pilot participant’s profile mapping in Phase 1, showing daily mood fluctuations and activity patterns over one week before the intervention. This visual tool helps both participant and occupational therapist recognize personal routines and occupations and set goals for the intervention’s next steps.

Building on these insights from Phase 1, in Phase 2, participants learn and begin practicing strategies for occupational and emotional balance. These include mapping weekly occupations, identifying personal patterns, and integrating balancing activities into their routines.

Phase 3 focuses on developing and applying a balanced daily schedule using strategies learned in the previous phase. Participants continue to monitor their routines, identify challenges, and practice and learn additional strategies to support sustained participation and well-being. Because the intervention supports expanding participation from the inpatient setting to the community, participants are encouraged to integrate real-life routines and habits in preparation for discharge.

Finally, Phase 4 centers on planning for the continued use of strategies after discharge. Participants develop a functional plan for transition to home and community environments, summarize effective strategies learned, and set future goals. The process concludes with a progress review and identification of personal anchors to support ongoing balanced routines and participation.

Table 3 provides a detailed overview of the themes and contents of each session.

## 4. Feasibility/Piloting

According to the MRC guidelines (Figure 1), the next step is to test the new intervention in a pilot study before embarking on full-scale research. These phases allow researchers to assess the study’s practical aspects, identify challenges in delivery or measurement, reduce uncertainties, and refine intervention procedures before formal evaluation [2]. The pilot phase of the MOBILE intervention was designed to evaluate the feasibility and acceptability of the intervention and its assessments among inpatients with MDD. Specifically, it examined the practicality of delivering the intervention in a hospital setting, the suitability of selected assessment tools, participant engagement and adherence, and any logistical or methodological challenges that require refinement before larger-scale research.

We conducted the pilot study with two patients hospitalized in the adult open ward at Shalvata Mental Health Center, Israel. The study received approval from the Faculty of Social Welfare & Health Sciences Ethics Committee, University of Haifa (approval no. 188/22), and the Shalvata Mental Health Center Institutional Review Board (approval no. 0016-22-SHA). Inclusion criteria were (a) age 18–65 years; (b) fluency in Hebrew; (c) confirmed diagnosis of MDD per DSM-5 by a treating psychiatrist; and (d) a Hamilton Depression Rating Scale score of at least 13. Exclusion criteria encompassed diagnosis of a psychotic spectrum disorder, neurological or developmental condition, chronic pain disorder, active suicidal ideation with plan and intent, or substance use within three months prior to participation. Both participants met all eligibility criteria and provided informed consent. Participation was voluntary, and no financial compensation or external incentives were provided for involvement or compliance with EMA protocols.

Both participants completed pre- and post-intervention assessments, which included standardized questionnaires and measures of occupational balance, daily functioning, resilience, and QoL. The assessment instruments—Occupational Balance Questionnaire (OBQ-11) [31], Life Balance Inventory (LBI) [32], Activity Card Sort (ACS) [33], Manchester Short Assessment of Quality of Life (MANSA) [34], and Connor-Davidson Resilience Scale (CD-RISC) [35]—were selected for their strong psychometric properties, sensitivity to short-term change, and widespread use in both clinical practice and research on MDD. This assessment battery was consistent with those used in our preliminary study (see [29] for further details). To evaluate treatment effectiveness, we included the Clinical Global Impression (CGI) [36], a tool widely utilized in clinical trials across a range of mental disorders, and the Canadian Occupational Performance Measure (COPM) [37], which is strongly supported as a valid outcome measure in both mental health research and real-world clinical settings [38].

Additionally, during the intervention, participants completed intensive daily monitoring for mood, activities, and resilience using the EMA protocol (five times daily for mood and activities and once daily for resilience, over 40 days). Prompts were delivered at random intervals within pre-specified time blocks during participants’ waking hours (7:00–23:00), in alignment with current best practices and recommendations for EMA research, thereby ensuring ecological validity and minimizing predictable response bias [39]. Data were subsequently used to measure dynamic changes in mood instability and daily functioning and quantitatively validate the reciprocal model underlying MOBILE through intra-individual analyses. Each participant received the full, personalized, eight-session MOBILE intervention protocol, delivered twice weekly within the ward setting.

Both participants completed all MOBILE intervention stages without dropout or adverse events, demonstrating high engagement. The assessment tools were well tolerated and appropriate for the clinical population. The intervention was delivered as planned: Both participants attended all eight sessions within the set timeframe. The EMA compliance was generally high, with an adherence rate of 71.8%. However, a slight decline in reporting toward the end of the study led to minor revisions in participant materials. The handbook and workbook were updated with more explicit reminders and clearer examples linking daily reports to functional outcomes. In addition, individual feedback based on participants’ EMA data was incorporated throughout the intervention to further support engagement and adherence. Notably, the feasibility findings presented in this work are based on a very small sample (N = 2) and thus must be interpreted as preliminary; nevertheless, the pilot demonstrated that the intervention and procedures were feasible and acceptable, justifying progression to the next research phase with minor refinement.

## 5. Discussion

This article presents the systematic development of the MOBILE intervention for individuals with MDD according to the MRC guidelines for complex interventions [1,2]. It offers an in-depth perspective on how theoretical models, empirical research, and iterative collaboration among stakeholders can be synthesized to create a novel OT intervention tailored to the specific needs, contexts, and lived experiences of people with MDD. Only a few publications explicitly described the phases and rationale underpinning the design of OT interventions for MDD, making this contribution especially significant for advancing evidence-based OT in mental health.

The MRC guidelines describe intervention development and evaluation as a dynamic, non-linear process, comprising four key phases: development, feasibility/piloting, evaluation, and implementation [1]. Recently, a newly introduced framework for developing complex interventions [2] expanded on the MRC guidelines. This framework supports researchers working alongside stakeholders in clarifying the main questions surrounding the intervention and in designing and conducting research that integrates diverse perspectives and appropriate methods. In the updated framework, a common set of six core elements guides each phase: considering context, developing and refining program theory, engaging stakeholders, identifying key uncertainties, refining the intervention, and addressing economic considerations [2]. These elements should be considered early and continually revisited throughout the process.

The MOBILE intervention is deeply embedded within its clinical and social context and considers the complexity and the unique demands of inpatient psychiatric settings. Acute psychiatric hospitalization can impose significant restrictions on individuals’ daily lives, limiting opportunities for daily functioning, participation, and achieving occupational balance [40]. At the same time, the structured environment of an inpatient ward, with its regular routines and schedules, provides opportunities for a gradual and supported return to everyday activities.

One strength highlighted during the MOBILE intervention’s development process is its inherent adaptability to different settings. Although originally designed for inpatient wards, it can be implemented in additional settings, such as participants’ homes, community centers, acute psychiatric units, and outpatient clinics. However, further refinement and adjustment are necessary to ensure its effectiveness across these diverse environments.

Another essential aspect in the development of the MOBILE intervention was the process of theory development and refinement. The intervention is grounded in an integrative synthesis of multiple conceptual frameworks, including occupation- and lifestyle-based models, mood- and function-based models, a client-centered and lived experience perspective, and findings from a preliminary empirical study as well as existing occupational therapy interventions. These elements were combined into the MOBILE model, which illustrates reciprocal relationships among depressive symptom severity, mood instability, and occupational patterns. This model offers a coherent theoretical rationale, empirically supported by EMA data, that structures the intervention logic and guides the selection of methods and anticipated pathways of change for participants.

Recent advances in neuroscience and dimensional psychopathology highlight the importance of grounding intervention models for depression within a broader biopsychosocial and multidimensional framework. The National Institute of Mental Health’s Research Domain Criteria (RDoC) approach emphasizes that MDD involves disturbances across continuous domains—from mood and cognitive functioning to social and neural processes—rather than discrete categories alone [41]. Integrating neurobiological insights, including altered neural circuitry and neuroplasticity allows for a more nuanced and person-centered understanding of both symptom patterns and functional impairment [3,42,43]. By embracing both approaches, future development and evaluation of the intervention can incorporate neurobiological outcome measures and enable precision-tailored rehabilitation, thereby aligning with contemporary research and enhancing both theoretical validity and clinical relevance.

The unique value of the MOBILE intervention lies in its integrative approach. Whereas other interventions often rely on retrospective feedback and primarily target isolated aspects such as time use or routine building, MOBILE delivers dynamic assessment and intervention tailored via real-time, ecologically valid feedback. Crucially, it simultaneously addresses both mood instability and daily functioning, emphasizing their reciprocal and balanced interplay. Its innovative nature is further strengthened by systematic monitoring of personalized participant goals and strategies, coupled with the incorporation of client and expert feedback, thereby optimizing ecological validity and relevance for mental health populations.

Integral to the development process is stakeholder engagement. The MOBILE intervention incorporates insights from expert occupational therapists and individuals with lived experience of MDD. This collaboration not only refines the intervention’s content and delivery, enhancing its clinical relevance and acceptability, but also ensures the program resonates with its intended users’ priorities and perspectives [2]. Such partnerships are recognized as fundamental in recovery-oriented programs [28]. Thus, in developing the MOBILE intervention, stakeholder feedback directly informed the adaptation of intervention components, clarified materials, and broadened its applicability across diverse care settings.

The fourth key element in developing complex interventions is identifying and addressing key uncertainties. Throughout the development process, the authors systematically identified and addressed potential challenges, particularly participants’ adherence to the EMA monitoring protocol, the feasibility of implementing the intervention in inpatient settings, and changes in participants’ health status over the course of the intervention. Given the small pilot sample, feasibility findings should be interpreted with caution. However, the piloting phase and stakeholders’ feedback allowed necessary protocol refinements and informed a research design tailored to the intervention’s aims and context. Future studies should replicate these findings with larger, more diverse samples to validate feasibility and further optimize protocol.

After identifying and managing key uncertainties, the fifth core element—intervention refinement—became especially important. This refinement is an ongoing and iterative process in which each phase of the MOBILE intervention’s development served as an opportunity for systematic adaptation and improvement. Such a dynamic process enhances the intervention’s fidelity while preserving the flexibility required for personalization and effective transfer from inpatient care to broader community contexts.

Finally, the sixth core element in the development process involves economic considerations. Although this study did not focus extensively on economic evaluation, it embeds early consideration of feasibility and resource use in its design. Integrating economic considerations early in intervention development supports methodological rigor and can help generate evidence that is more relevant for decision-makers in real-world clinical settings. These considerations provide an important foundation for future research on cost-effectiveness, which can facilitate efficient resource allocation and ensure the long-term sustainability of complex interventions [44].

This article focuses on the MOBILE intervention’s development; a limitation is the absence of a detailed description of its evaluation and implementation phases. These two phases are critical for establishing whether the intervention achieves its intended outcomes and understanding how it can be effectively adopted, adapted, and sustained in real-world clinical practice [1,2]. Future research evaluating the MOBILE intervention should be designed with careful consideration of the target population and appropriate measures to assess both intervention outcomes and dynamic changes throughout the intervention process. Study designs should integrate standard single-time-point assessments of participation and occupational balance with continuous real-time assessment methods such as EMA. Utilizing a tool like EMA for data collection represents a novel approach that can capture dynamic changes in mood, occupational patterns, and real-life experiences among participants. It increases the findings’ ecological validity while enabling a more nuanced understanding of the intervention’s effects [39].

To advance the evaluation and implementation of the MOBILE intervention, the next step will involve adopting a single-case experimental design (SCD) incorporating repeated EMA and standardized functional measures. This approach enables rigorous, individualized assessment of intervention effects in clinical settings. Drawing from empirical and theoretical foundations, it is hypothesized that participation in the MOBILE intervention will enhance mood stability, occupational balance, resilience, and QoL among individuals with MDD. Single-case experimental designs provide a flexible and methodologically robust framework for tracking intra-individual changes over time, allowing each participant to serve as their own control and supporting more accurate causal inferences regarding intervention effectiveness, especially in heterogeneous or small samples [45]. The compatibility of this methodology with EMA facilitates frequent, real-time assessment of mood and functional outcomes, capturing patterns and changes that may be overlooked by traditional retrospective measures [46].

Single-case experimental designs are particularly advantageous in the context of complex mental health and rehabilitation interventions, as they make it possible to closely monitor variability and individual responsiveness throughout the intervention process. Incorporating SCD and EMA in future research could not only improve the scientific rigor of outcome evaluation but also align with current best practices in studying complex psychosocial interventions, facilitating gradual refinement, clinical implementation, and broader knowledge translation [47].

The literature emphasizes that translating interventions from controlled research settings into everyday clinical practice is a multifaceted process. Organizational culture, stakeholder engagement, and ongoing adaptation to contextual realities influence such implementation [48]. Thus, knowledge translation strategies have been shown to shift professional attitudes, enhance clinical skills, and increase access to evidence-based rehabilitation, as demonstrated in rehabilitation settings and specifically within OT for mental health and cognitive impairment [49,50]. These strategies have also been associated with better long-term integration of evidence-based practices and improved outcomes over time for complex interventions [51].

An important facilitator for the MOBILE intervention’s future implementation is the in-depth and systematic analysis of its fundamental components, as exemplified by the rehabilitation treatment specification system (RTSS) [52]. The RTSS constitutes a structured, theory-informed framework that precisely delineates intervention targets, active therapeutic ingredients, and the hypothesized mechanisms of action leading to change. Integrating the RTSS into future implementation processes will enhance conceptual clarity, ensure consistent and transparent delivery across diverse clinical environments, and facilitate adaptation while preserving intervention fidelity [53]. This approach not only addresses the ‘black box’ challenges of complex rehabilitation interventions but also ensures that the MOBILE intervention remains replicable, adaptable, and aligned with its core therapeutic objectives as it is scaled and adopted in varied practice settings.

Additional considerations and actions in the implementation process should include establishing continuous feedback mechanisms to monitor barriers and progress, considering real-world policy factors, and addressing economic aspects like resource allocation to ensure the intervention’s large-scale delivery and sustainability [2,48].

In summary, this article provides a detailed account of the development of the MOBILE intervention in accordance with MRC guidelines. The MOBILE intervention is specifically tailored to address the nuanced interplay between mood instability and occupational imbalance, two central challenges in MDD that directly affect QoL. This article addresses a critical gap in the field of OT interventions within mental health and demonstrates the value of a systematic, contextually sensitive, and theory-informed development process. Such an approach paves the way for rigorous evaluation and widespread implementation of client-centered and lived experience-based interventions for individuals with MDD.

## Figures and Tables

**Figure 1 healthcare-13-02667-f001:**
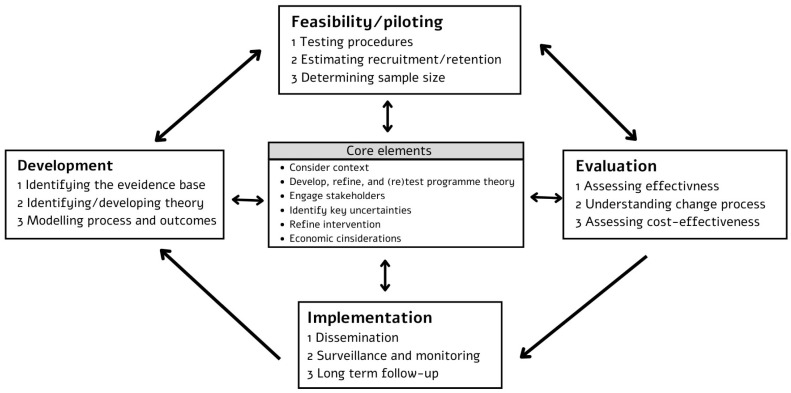
Framework and key elements for developing and evaluating complex intervention [1,2].

**Figure 2 healthcare-13-02667-f002:**
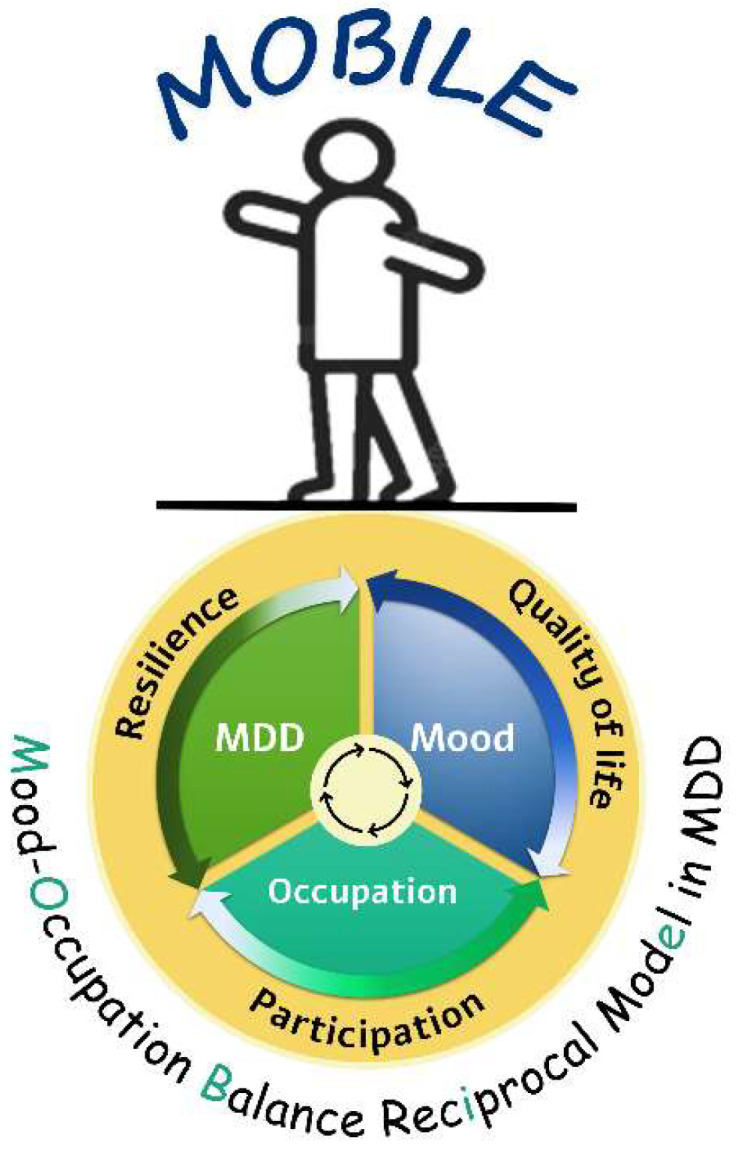
The mood–occupation balance reciprocal (MOBILE) model.

**Figure 3 healthcare-13-02667-f003:**
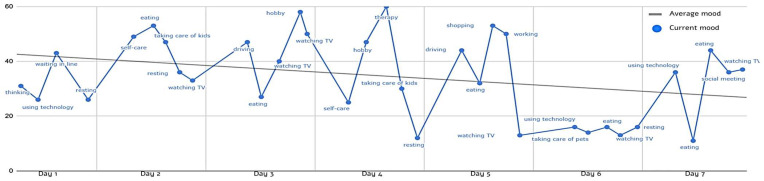
EMA-based mood and activity during the first week of baseline monitoring (average mood = 34.27).

**Table 1 healthcare-13-02667-t001:** Key domains and guiding questions for stakeholder feedback.

Domain	Guiding Questions
Clarity and understanding	Are the explanations in the protocol and participant manual clear and understandable? Is the language suitable and considerate of the target audience?
Intervention structure	Is the protocol organized and evidence-based? Are components clearly described and adaptable?
Relevance and content	Are the topics and tasks proposed in the program relevant to your challenges? Are important topics missing?
Implementation	Are procedures clear and easy to follow? Are barriers addressed? Are participation requirements reasonable and feasible?
Flexibility and personalization	To what extent does the program allow flexibility and adaptation to the participant’s personal goals?
Overall opinion	What are the program’s strengths and limitations? What improvements are recommended? What parts of the intervention are most helpful? Are there parts that may be intimidating or disruptive to participation?

**Table 2 healthcare-13-02667-t002:** Application of underlying theories and models in the MOBILE intervention.

Underlying Theory/Model		Key Component	Application in the MOBILE Intervention
Occupation- and lifestyle-based models	Time-use model	Allocation of time across activities Meaning and variety of occupations Effect of diverse contexts on participation	Map routines and time use Monitor with time-use diaries Reflect on participation patterns Adjust routines for occupational balance and meaning
	Lifestyle balance model	Pattern of daily occupations Alignment with values, skills, meaning, and interests	Identify meaningful activities Support congruence between routines and personal circumstances
Mood- and function-based models	Daily function homeostasis	Mapping daily function Raising awareness of occupational patterns through structured questions	Design personal profiles of daily activities Identify strengths and needs for change Use strategy for function regulation
	Mood homeostasis	Mood regulation through activity selection Promoting awareness of factors affecting daily occupational choices	Monitor mood and function Reflect on daily states Promote adaptive strategies for mood management
	MOBILE model	Reciprocal relationships between depression severity, mood instability, and occupation Significance of ecological experience for understanding daily functioning challenges	Address all three domains simultaneously Raise awareness of the relations between key components EMA-based mood and activity monitoring Personalized goal-setting and strategy use
Client-centered and lived experience approach		Collaboration Individual needs, values and goals Personal meaning and context Stakeholder-driven involvement	Personalize goal-setting Tailor intervention and strategies Monitor real-life experiences Adapt flexibly to the participant’s personal circumstances
Multicontext approach		Optimizing functional cognition Awareness and strategy use Transfer and generalization across contexts	Promote self-awareness and strategy use Apply strategies in varied real-life situations

Note. EMA = ecological momentary assessment; MOBILE = the mood–occupation balance reciprocal model for major depressive disorder.

**Table 3 healthcare-13-02667-t003:** Themes and content of the eight MOBILE intervention sessions.

Phase	Session	Theme	Session Goal	Content
1: Initial steps	1	First step toward balance	Orientation & goal setting	Review assessment results (occupational balance, participation, mood tracking), introduce participant’s personal functional profile, identify individual functional goals for ongoing work
	2	Building the foundation	Theoretical integration	Introduce core concepts and their impact on daily functioning, connect theoretical concepts to participant’s daily experiences, identify real-life situations in participant’s daily life expressing the connection between occupational balance and mood
2: Tools for the journey	3	Small steps for balance (occupational balance strategies)	Strategy acquisition	Map weekly occupations and create a personal occupation map, raise awareness of occupation types and occupational perspectives, identify personal occupational patterns, learn strategies for occupational balance, and initiate one change in daily routine.
	4	Steps toward emotional balance (emotional regulation strategies)	Strategy acquisition	Map emotional states throughout the week, increase awareness of the connection between functioning and emotional states, learn strategies for emotional regulation, identify and integrate balancing activities into the daily routine.
3: Taking action and building routine	5–7	Building a balanced routine	Implementation & practice	Continue to monitor the weekly functional schedule, identify challenges, learn and integrate new strategies, incorporate strategies into daily planning, expand functioning to natural environments, build a balanced weekly routine promoting broader participation in daily occupations
4: Planning ahead	8	Looking forward	Expansion & maintenance	Plan to expand functioning to home and community settings, develop functional plan for hospital discharge, summarize effective strategies for change and maintaining a balanced daily routine, review intervention process and set future goals

## Data Availability

Data is not available as it contains sensitive and confidential information, which survey respondents were assured would remain private and not be shared.

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
