# Peer review of "From Theory to Practice: Developing the MOBILE Intervention for Adults with Depression"

_healthcare, 2025, doi:10.3390/healthcare13212667_

Round 1
Reviewer 1 Report
Comments and Suggestions for Authors
The manuscript presents the development of the MOBILE intervention, a client-centered occupational therapy program designed for adults with major depressive disorder.
Consider the following points:
1. The last sentence of the abstract is not a conclusion: "This article provides a comprehensive account of development procedures to support future evaluation, implementation, and integration into OT mental health practice."
2. The paper presents the contend of Section 2 (line 67), but it does not present the paper organization. Provide it.
3. Line 142: "The following sections describe the key theories..."; following paragraphs?
4. There is no section 3 in the manuscript.
5. Table 2 is a core part of the paper. The "application in the MOBILE intervention" could be better described/explored (figures?) in the manuscript. For example, it's hard to get "Design personal profiles of daily activities".
6. What about inserting screenshots to better present the 'Content' in Table 3?
7. Main issue (related to the previous 2 comments): the MOBILE model is well-developed conceptually, but the mapping between theoretical constructs and the specific intervention techniques could be described with greater precision (beyond what is summarized in Tables 2 and also 3).
Reviewer 2 Report
Comments and Suggestions for Authors
I congratulate the authors on their choice of topic and the study they have developed.
Considerations:
The abstract does not cover the objective of the study.
The introduction refers twice to the objective of the study, and the two are not exactly the same.
The theory is very well explained, as is the pilot phase.
The introduction begins with the objective, and later on the objective is referred to again, and the two are not exactly the same.
The bibliography is recent, with most references less than five years old.
It does not mention how the two participants were selected, nor the inclusion criteria or necessary characteristics.
Suggestions:
Clarify the objective of the study and refer to it consistently.
Clarify how the participants were selected and what the inclusion criteria were.
Better organise the structure of the introduction.
Reviewer 3 Report
Comments and Suggestions for Authors
This manuscript presents the systematic development of the MOBILE intervention, a novel occupational therapy (OT) program tailored for adults with major depressive disorder (MDD). The work is grounded in the Medical Research Council (MRC) framework for complex interventions and integrates theoretical models, ecological momentary assessment (EMA) data, and stakeholder input. The article is valuable in providing a structured description of the development process of a client-centered and theory-driven intervention, which is still rare in OT literature for mental health. The paper is well organized, and the intervention appears innovative, relevant, and feasible. Nevertheless, some issues require further clarification, elaboration, and strengthening to improve the manuscript’s rigor and impact.
Major Comments:
- The pilot study includes only two participants, which limits the ability to draw meaningful conclusions on feasibility beyond anecdotal observations. The authors should clearly acknowledge this limitation earlier in the manuscript (perhaps already in the abstract and introduction) and discuss how it constrains generalizability.
-
Although the manuscript cites relevant OT interventions (ReDO, BEL, Action Over Inertia), the comparison remains somewhat descriptive. A more critical positioning is needed: what specifically differentiates MOBILE from these existing approaches? Is the added value mainly the integration of mood instability monitoring, the use of EMA, or the reciprocal model?
- EMA is central in the rationale and piloting, but the manuscript does not fully explain how EMA data will be systematically used in later phases of evaluation (e.g., to tailor sessions, measure change, or validate the reciprocal model). Authors should clarify this point.
-
The discussion acknowledges that evaluation and implementation are missing, but the paper could go further in outlining concrete next steps. What kind of study design (e.g., single-case experimental designs, RCTs) is most feasible? What are the authors’ hypotheses regarding expected outcomes? This would make the paper more forward-looking.
- Additionally, while the manuscript is rich in occupational and psychosocial models, the discussion could be strengthened by briefly considering potential neural substrates or mechanisms underlying mood–function interactions. Recent reviews and empirical studies have emphasized the importance of linking behavioral change to neural processes, which could help situate the MOBILE intervention within a broader biopsychosocial framework (e.g., doi.org/10.1016/j.neubiorev.2025.106273; doi.org/10.1038/s41392-024-01738-y; doi.org/10.1038/s41598-025-13185-y; doi.org/10.1016/j.neubiorev.2025.106273).
-
The description of assessment tools is somewhat limited. Given the focus on participation, resilience, and QoL, maybe it would be useful to justify why specific instruments were chosen, and whether they are sensitive enough to capture short-term intervention effects.
Minor Comments
- At times, the manuscript alternates between “mood instability,” “mood homeostasis,” and “emotional regulation.” While related, these are not interchangeable terms.
- Figure 2 (MOBILE model) is central, but the visual is not very detailed. A clearer schematic showing arrows/feedback loops could improve communication of the reciprocal relationships. Similarly, Table 3 could be reformatted to highlight progression across phases more clearly.
- The role of “experts by experience” is a strong point of the study. However, it is described rather briefly. Could the authors expand slightly on how their feedback differed from that of occupational therapists, and whether their suggestions led to concrete protocol changes?
- The ethics section is clear, but it might be worth specifying whether participants in the pilot received compensation or incentives, particularly for EMA compliance.
- Minor grammatical inconsistencies are present (e.g., “participants are encouraged to integrate real-life routines…” could be rephrased for smoother style). A final language polish would be beneficial.
The quality of English is generally good, clear, and professional. Minor issues are present, mainly related to consistency (e.g., switching between “mood instability,” “mood regulation,” and “mood homeostasis”), occasional long and dense sentences that could be streamlined, and small grammatical inconsistencies.
Round 2
Reviewer 1 Report
Comments and Suggestions for Authors
The authors addressed all my concerns.
Author Response
We thank you for your thoughtful feedback .
Reviewer 3 Report
Comments and Suggestions for Authors
Dear Authors,
Thank you for your thoughtful revisions and for addressing the previous comments on your manuscript, "From Theory to Practice: Developing the MOBILE intervention for Adults with Depression." The paper provides a clear, systematic, and much-needed account of a complex intervention's development, which is a significant contribution to the field of occupational therapy in mental health. The adherence to the MRC framework is commendable and serves as an excellent model for future research.
I have a few minor suggestions to further strengthen the manuscript before publication, including a point from the previous review that I believe warrants further consideration.
1. In the discussion of the pilot study, authors note that a slight decline in EMA adherence led to "minor revisions to the participant handbook accompanying the intervention". The manuscript would be strengthened by briefly specifying the nature of these revisions (e.g., adding more explicit reminders, simplifying the language, providing clearer examples, etc.). This would offer a more concrete example of the iterative refinement process guided by the MRC framework and provide valuable insight for other researchers developing similar protocols.
2. The pilot study section mentions that participants completed EMA monitoring "five times daily for mood and activities and once daily for resilience, over 40 days". While concise, adding a sentence about the timing of these prompts (e.g., "at random intervals within pre-specified time blocks" or "at fixed times") would provide greater methodological clarity for replication and assessment by the reader.
These are minor suggestions intended to further enhance an already strong and valuable contribution. I commend the authors again on the rigorous and transparent development of the MOBILE intervention.
